# Protocol of mixed-methods assessment of demographic, epidemiological and clinical profile of decentralised patients with cancer at Nelson Mandela Academic Hospital and Rob Ferreira Hospital, South Africa

Wezile Chitha,[1] Zukiswa Jafta,[1] Onke R Mnyaka  ,[1,2] Danleen Hongoro,[1] Lizo Godlimpi,[3] Buyiswa Swartbooi  ,[1] Natasha Williams,[1] Christopher Zungu,[1] Lazola Buthi,[4] Sibulelo Kuseni,[4] John Nasila,[1,5] Siyabonga Sibulawa,[1] Olona Giwu,[1] Awam Mavimbela,[1] Vivien Essel,[1] Sikhumbuzo A Mabunda  [6,7]

VE and SAM are joint senior authors.

For numbered affiliations see end of article.

**Correspondence to**
Onke R Mnyaka;
onkemnyaka@gmail.com

## ABSTRACT

**Introduction** Cancer is the second leading cause of death globally. However, cancer care services are often concentrated in urban centres. Two of South Africa's hospitals have decentralised cancer care delivery since February 2018 and August 2019, respectively. This study aims to describe the demographic, epidemiological and clinical profile of various cancers at Nelson Mandela Academic Hospital (NMAH) and Rob Ferreira Hospital (RFH), in South Africa's Eastern Cape and Mpumalanga provinces, respectively.

**Methods and analysis** This study will be conducted in the Eastern Cape and Mpumalanga provinces. A mixed-methods study design will be undertaken to gain insight on the characteristics of randomly sampled patients who are treated for cancer at NMAH and RFH between 1 March 2018 and 28 February 2022. A validated, researcher-administered survey questionnaire will be used to assess demographic characteristics, and prevalence of different cancers among patients. Concurrently, a document review will be undertaken on patients with cancer using a patient registry to ascertain the duration of diagnosis, type of cancer(s), management plan and patient survival time. STATA V.17 will be used for data analysis. The Shapiro-Wilk test will be used to explore the distribution of numerical variables. The $\chi^2$ or Fisher's exact tests will be used depending on the value of the expected frequencies to compare categorical variables. Kaplan-Meier survival estimates will be used to determine the survival time. Hazard ratios will be used to determine the predictors of death. The level of statistical significance will be set at p value ≤0.05. The 95% CI will be used for the precision of estimates.

**Ethics and dissemination** Ethics approval was obtained from the Human Research Ethics Committees of the University of the Witwatersrand (M210211) and Walter Sisulu University, South Africa (Ref: 040/2020). Findings will be reported through peer-reviewed journal(s), presentations at conferences and at partner meetings.

## Strengths and limitations of this study

► To our knowledge this is the first study to formally report on decentralisation of cancer care services in South Africa.
► Triangulation of designs compensates for the potential limitations of a single design and thus provide more insight on cancer care delivery models in the selected study sites.
► The ambi-directional cohort design does not only enable the assessment of the survival time and predictors of death but also enables the retrospective and prospective follow-up of patients and thus understand their care plans better.
► The study could be limited by the quality of data or poor information systems thus resulting in missing data.

## INTRODUCTION

Cancer is considered to be the number two cause of death globally, accounting for an estimated 9.6 million deaths.[1 2] Africa has the second lowest rate of cancer-related deaths contributing 7.1% to the total cancer deaths globally.[3] Cancer is expected to continue to rise as part of the epidemiological transition globally, further straining limited healthcare resources.[1] Signs of this prediction have become more visible with rapidly growing global cancer incidence and mortality rates.[3]

Approximately a third of cancer deaths are due to behavioural and dietary risks, such as, high body mass index, low fruit and vegetable intake, lack of physical activity, tobacco use and alcohol use.[4] For example, smoking is the most common preventable cause of

BMJ

premature mortality worldwide but it contributes to almost 30% of all cancers in high-income countries.[5] These risk factors are preventable[6] and may be substantially reduced through adjustments in lifestyle.[2]

In South Africa, cancer is a growing national health and socioeconomic concern.[6] According to the International Agency for Research on Cancer in 2020, there were 108 168 new cancer cases in South Africa, bringing the risk of developing cancer before the age of 75 years to 20.7% (23.6% male and 18.7% female).[7] In the same year, 56 802 deaths were reported, bringing the risk of dying from cancer before the age of 75 years to 11.8% (13.9% male and 10.4% among female).[7] The increasing incidence and mortality rates present a huge challenge to the affected patients and their families especially those who have limited access to care.[6]

In 2016, the top five cancers affecting women in South Africa were breast cancer (27.1%), cervical cancer (18.7%), colorectal cancer (6.3%), lung cancer (4.9%) and cancer of the uterus (3.9%); while the top five cancers affecting men were prostate cancer (25.8%), lung cancer (12%), colorectal cancer (7.3%), Kaposi sarcoma (4.9%) and non-Hodgkin's lymphoma (4.1%).[3] The Nelson Mandela Academic Hospital (NMAH), Eastern Cape province and Rob Ferreira Hospital (RFH), Mpumalanga province embarked on a decentralised model of cancer care delivery in February 2018 and August 2019, respectively. Decentralisation refers to making cancer services available in certain hospitals that previously did not have any cancer care service provision, such as our two study sites. In this way, patients can access quality cancer services closest to where they live, health professionals will be able to screen and diagnose early, unnecessary delays to treatment will be reduced and patients will get quality palliative care closer to their families. A positive effect of this proposition is that patients' and families' healthcare-related out of pocket costs will be reduced.

The two hospitals aim to establish centres of excellence in cancer care, a network of cancer care satellite sites at district hospital level and community-based cancer care services. Decentralisation is meant to be achieved in four different phases. The first phase entailed decentralisation of chemotherapy services from Frere Hospital (East London) and Inkosi Albert Luthuli Hospital (Durban, KwaZulu-Natal province) to NMAH in February 2018 for patients from the OR Tambo district (the district where NMAH is located), and three other neighbouring districts for patients in the Eastern Cape province. For Mpumalanga province, all chemotherapy services were decentralised to RFH from Steve Biko and Kalafong Hospitals in Gauteng province in August 2019. The two hospitals (NMAH and RFH) were assisted with the hiring of radiation and medical oncologists, oncology-trained professional nurses, pharmacists, social workers, clinical psychologists, ultrasound and mammogram technicians, and administrators. Equipment includes a spirometer, mammogram, ultrasound, colposcopy, and a large loop excision of the transformation zone machines in both hospitals.

The second phase (current phase) entails decentralisation of chemotherapy services further from NMAH to four regional hospitals and one district hospital in the Eastern Cape. However, only one of the five hospitals has been fully decentralised from February 2021, the other four hospitals have achieved partial decentralisation with the procurement of equipment. For Mpumalanga, the second phase entails decentralisation of chemotherapy services from RFH to Witbank Hospital for patients from two of their three districts from May 2021.

The third phase will be full decentralisation of radiotherapy services from Frere Hospital to NMAH in the Eastern Cape and from Steve Biko and Kalafong Hospitals to RFH, then Witbank Hospital in Mpumalanga. The fourth and final phase is the strengthening of district hospitals and community-based services to manage aspects of cancer effectively from screening, diagnosis, treatment and palliative care support. This phase will also increase the pool of oncology-trained nurses and medical officers at primary care and district hospital level. These latter two phases are still outstanding. Figure 1 summarises the timeline of the current decentralisation process.

Evidently, the current decentralised model of care is only limited to patients requiring chemotherapy. It is hoped that the continued implementation of the decentralised model of cancer care will improve patient experience and quality of cancer care and reduce morbidity and mortality. However, data on patient demographic characteristics, epidemiological and clinical profile of various cancers are lacking in these two hospitals. This study therefore seeks to conduct an assessment of demographic, epidemiological and clinical profile of various cancers in NMAH and RFH. Furthermore, the study aims to describe the current process and its benefits/challenges, with hopes of expanding 'decentralisation of care' in terms of services offered and number of decentralised hospitals.

## Significance

South Africa's cancer services are generally urban based and located in tertiary and quaternary health centres with an underdeveloped cancer service platform at district hospital and primary care levels. This means that patients needing cancer care have to travel long distances to big cities/towns in order to access basic cancer care. This creates gaps in access and quality of cancer care delivery between urban areas and rural areas. Decentralisation is a result of operational observations (to our knowledge there is no formal research that was done) such as long waiting times, delayed presentations, late diagnosis, patient complaints on travelling and out of pocket costs, and so on. It is therefore envisaged that this study will provide insight on the distribution and types of cancers in areas where there is currently an underestimation of the burden of disease and as a result incorrect understanding of the levels of risk within the local populations. Moreover,

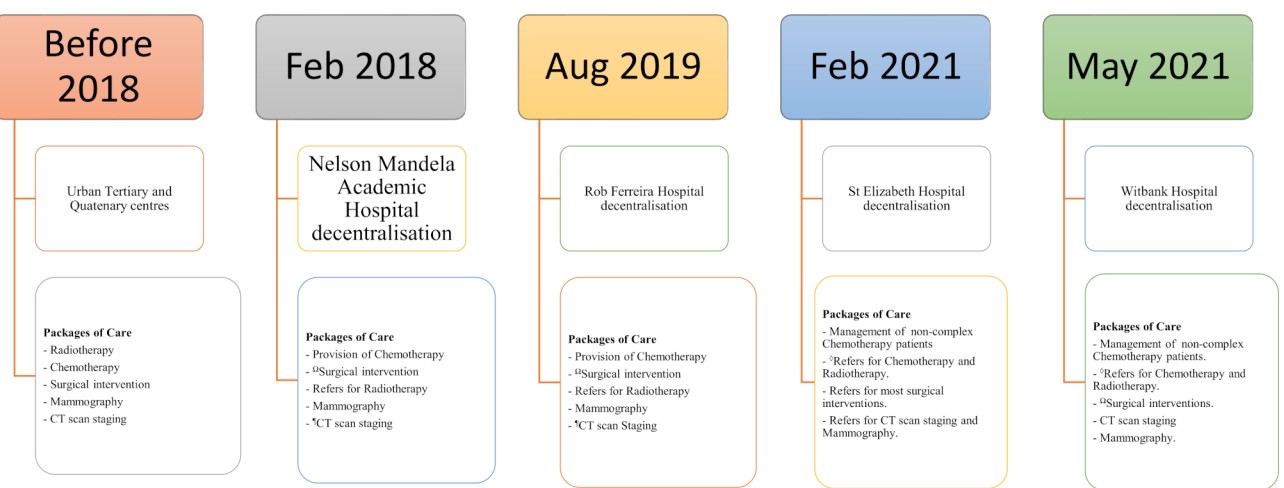

**Figure 1** Timeline of the decentralisation process. ΩPerformed most surgical interventions even before decentralisation. ¶Performed this function before decentralisation. ◊No provision of chemotherapy in the hospitals due to lack of lamina flow.

establish the extent of the problem in both health facilities and possibly justify the need for decentralisation of cancer care services and help inform cancer preventive strategies in South Africa and other similar settings.

## Objectives
1. To describe the sociodemographic characteristics of patients diagnosed with cancer in the selected hospitals in the Eastern Cape and Mpumalanga provinces, South Africa.
2. To determine and compare the incidence rate and prevalence of different types of cancer in the selected hospitals.
3. To determine and compare the geographic distribution of cancers in the Eastern Cape and Mpumalanga provinces of South Africa.
4. To determine the gaps between symptom development, first presentation at a health institution, first cancer diagnosis, referral for definitive management and initiation on treatment of patients diagnosed with cancer in South Africa's Eastern Cape and Mpumalanga provinces.
5. To determine the comorbid conditions of patients with a cancer diagnosis in South Africa's Eastern Cape and Mpumalanga provinces.
6. To determine the survival time of patients diagnosed with cancer in South Africa's Eastern Cape and Mpumalanga provinces.

## METHODS AND ANALYSIS
### Research design
This study will use a quantitative approach with a triangulation of a descriptive, exploratory cross-sectional and a longitudinal cohort design to answer the study objectives. The triangulation of designs compensates for the potential limitations of a single design. This study forms part of a bigger but yet to be published research project titled: 'Exploring the feasibility, implications and outcomes of decentralising cancer care delivery in the Eastern Cape and Mpumalanga provinces of South Africa'.

Information will be sourced through two quantitative substudies, a cross-sectional survey with patients with cancer and an ambi-directional cohort document review. Below is a brief description of the two substudies.

### Substudy 1: quantitative cross-sectional study
A quantitative survey questionnaire will be administered on patients to assess demographic characteristics, prevalence of different cancers in selected hospitals and compare geographic distribution of cancers in the Eastern Cape and Mpumalanga provinces.

### Substudy 2: quantitative ambi-directional cohort study design
Using the registry of patients with cancer (the registry is similar to a clinic logbook, but it is in an electronic form) used in the study sites, a document review will be carried out on patients with cancer to ascertain the duration of cancer diagnosis, type of cancer(s) and the duration of survival since admission to the oncology clinic (survival time). Figure 2 shows the ambi-directional component of the study. Table 1 summarises the two substudies.

### Study setting
The study is located in two rural provinces with a high degree of underdevelopment and marginalisation, namely Eastern Cape and Mpumalanga provinces in South Africa.[8] Generally, Eastern Cape and Mpumalanga provinces are characterised by lack of the necessary infrastructure, resources and expertise to provide quality, safe and accessible radiotherapy, chemotherapy, palliative care services and surgical oncology services. Patients from rural communities, who generally cannot afford private healthcare and are dependent on state health services for cancer care, are compelled to travel long distances to the urban-based tertiary or quaternary cancer care centres in order to access cancer care. The study will be conducted in two hospitals, NMAH in Mthatha, Eastern Cape province

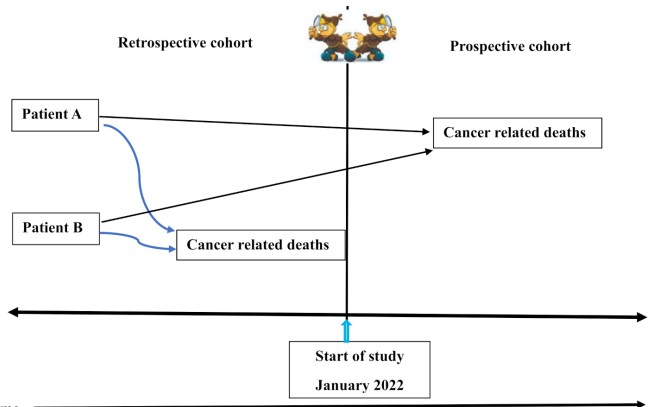

**Figure 2** Ambi-directional cohort study for substudy 2.
*Note*: Data collection will commence in January 2022 till the end of September 2022. Record review of patients diagnosed with cancer from 1 March 2018 will be done. Patients will be followed up for survival time either retrospectively (cancer-related deaths between 1 March 2018 to January 2022), or prospectively (cancer-related deaths from January 2022 until the study's end date, 30 September 2022).

and RFH in Mbombela, Mpumalanga province. NMAH is 1 of 10 central hospitals in South Africa and is the only one that is located in a rural area. This level of care is meant to be a quaternary level of care. RFH is a tertiary level of care hospital.

These are two hospitals in their respective provinces that refer their patients to seek quality cancer care in hospitals which are further away. At times, it takes patients up to 3 days of travelling when attending to their cancer care appointments. For example, patients with cancer from Mpumalanga's RFH travel more than 400 km to the country's capital city, Pretoria. While patients with cancer from the Eastern Cape's NMAH travel more than 200 km to East London to access quality cancer care at an urban-based tertiary hospital. An anomaly, as NMAH is statutorily a level of care higher than a tertiary hospital.

## Population and sampling

Purposive sampling was used to select the study hospitals. The hospitals were selected based on their levels of care, gazetted specialist packages of care and concerns about the existing package of cancer care services, and because they are currently implementing a decentralised model of cancer care delivery. Furthermore, the two hospitals aim to establish centres of excellence in cancer care, a network of cancer care satellite sites at district hospital level and community-based cancer care services.

A triangulation of approaches will be used to select study participants from the two hospitals.

### Substudy 1: quantitative cross-sectional study (patients)

Systematic random sampling of patients visiting the oncology clinics' outpatient's departments will be conducted by approaching every fifth patient in the queue until the sampling size has been reached. A total combined sample size for the two hospitals will be calculated using the equation, $n = \frac{p(1-p)z^2}{d^2}$ for a one-sided 95% CI and a 5% significance level (z=1.96). Because the proportion (p) of patients with cancer who are seen in the respective hospitals is not known, this (p) will be set at 50% and the margin of error (d) will be set at 5%. This thus yields a total minimum sample size of 385. To factor for data entry errors, a further 10% (39) will be added to yield a desired sample size of 424 participants for the two sites. Participants will then be recruited proportionally to yield a sample size of 212 patients per site.

### Substudy 2: quantitative ambi-directional cohort (document review)

Information will be extracted from the patient registry (the registry is similar to a clinic logbook, but it is in an electronic form) to respond to the questions on the extraction tool. All patients under the care of the unit at any stage between 1 March 2018 and 28 February 2022 will be included.

| Table 1 | Research methods summary | | |
|---|---|---|---|
| Substudy | Study design | Objectives | Analysis |
| 1 | Cross-sectional study | ► Describe sociodemographic characteristics of patients.<br>► Determine and compare the geographic distribution of cancers.<br>► Determine cancer disease progression.<br>► Determine comorbid conditions of patients with cancer. | ► Frequency tables, percentages and graphs to summarise categorical variables.<br>► Mean, SD and range to summarise normally distributed numerical variables; or Median and IQR to summarise skewed numerical variables.<br>► $\chi^2$ statistics or Fisher's exact test to compare categorical variables between groups. |
| 2 | Ambi-directional cohort study (document review) | ► Determine and compare the incidence rate and prevalence of different types of cancer in the selected hospitals.<br>► Determine the comorbid conditions of patients with a cancer diagnosis.<br>► Determine the survival time of patients diagnosed with cancer. | ► Parametric and/or non-parametric tests to compare numerical variables between groups.<br>► Kaplan-Meier survival estimates, for survival time.<br>► Hazard ratios for predictors of death. |

## Data collection

A multimethod approach to data collection will be adopted to get a comprehensive picture on cancer in the selected hospitals in terms of demographic distribution of cancer, socioeconomic characteristics, prevalence, duration of diagnosis, and so on. This approach will also compensate for the potential limitations of a single data collection method and to triangulate the data as a means of checking the consistency of the study findings.

### Substudy 1: quantitative cross-sectional study (patients)

The aim of this survey is to assess socioeconomic demographic characteristics of patients with cancer, prevalence of different cancers in selected hospitals and compare geographic distribution of the different cancers from the end-user's perspective. This substudy will adopt and use a standardised and validated quantitative survey tool (online supplemental appendix A) to collect data from patients. The survey tool for patients has 51 questions developed through literature review and whose content validity was reviewed by three experts (one occupational medicine specialist, a public health medicine specialist and an oncologist). The questionnaire asks about the patient demographic profile and epidemiological and clinical profile of various cancers. To test and ensure the effectiveness, the survey tool was also piloted among six patients in the two hospitals. Once the pilot study was done, all necessary adjustments were made to the data collection tool, thus ensuring that all questions will enhance the validity and reliability of the study findings. On clarity, there was 100% agreement among all three experts. On relevance only one of the three experts scored one question as irrelevant to result in an average score content validity index of 0.99 (99%), which still renders the tool valid. Online supplemental appendix C shows the experts' scoring in detail. This questionnaire has been translated into the local languages (isiXhosa, siSwati, and isiZulu) to accommodate participants who might not be comfortable with English.

### Substudy 2: quantitative ambi-directional cohort study (document review)

Using a data extraction tool (online supplemental appendix B), a document review will be conducted in addition to the survey questionnaire. The main aim of the document review is to ascertain information which could not be captured or verified from the survey questionnaire, including, duration of cancer diagnosis, types of cancer and survival time of patients with cancer from the date of diagnosis.

## Data management and analysis

Quantitative data analysis will include capturing survey data into Microsoft Excel Office 2016 and exporting the data into STATA V.17 (STATA Corp, College Station, Texas, USA) for analysis. Some descriptive and categorical data will be compared using frequencies, percentages and graphs. Numerical data will be explored for normality using the Shapiro-Wilk test.[9] If normally distributed, the mean, range and SD will be used. If not normally distributed, then the median and IQR will be used. The Wilcoxon rank-sum or an appropriate two-sample t-test will be used to compare the mean or median age of patients with cancer by cancer type and between the two sites depending on the normality of the distribution of age and/or the equality of variances. A test for the equality of variances will be performed before use of the two-sample t-tests, if numerical variables are normally distributed. The two-sample t-test for independent samples will be carried out if variances are equal, and Satterthwaite's modified t-test used if the variances are not equal. The $\chi^2$ or Fisher's exact tests will be used depending on the value of the expected frequencies. If expected frequencies are <5 in binary comparisons or if any one cell of a larger comparison has an expected frequency of <1 or more than 20% of the cells of nominal categorical comparisons have an expected frequency of <5, then the Fisher's exact test will be used. Kaplan-Meier survival estimates will be used to determine the survival time. Hazard ratios will be used to determine the predictors of death. The level of statistical significance will be set at p value ≤0.05. The 95% CI will be used for the precision of estimates. Through triangulation, the analysis of the survey will seek to reproduce the cancer prevalence, epidemiological profiles and demographic characteristics determined in substudy 1. These should be similar as determined by statistical methods (95% CI and two-sample test of proportions).

## Limitations

The study could be limited by the quality of data or poor information systems thus resulting in missing data. The assessment of the data at three time points will allow for the reduction of missing data as we will note suspicious entries and/or missing data and request for assistance with correction/filling of the missing data using the patient records or confirming from patients and/or family members. Missing data will be analysed using complete case analysis. The main outcome (cancer-related deaths) could be limited by the absence of a mechanism linking to the death certificate, and/or autopsy. Every unspecified natural cause of death will therefore be considered to be related to the cancer as a direct or associated cause of death.

## Patient and public involvement

The planning of the cancer service expansion involved community representatives through hospital boards in workshops and meetings. Patients will be informed of the study at all stages through consultations and public notices in the study sites.

## Ethics and dissemination

Ethics approval was obtained from the Human Research Ethics Committees of the University of the Witwatersrand (M210211) and Walter Sisulu University, South Africa

(040/2020). Site access approval has been obtained from the Provincial Health Research Committees of the Eastern Cape (EC_202010_012) and Mpumalanga (MP_202011_002) provinces, respectively. The study will abide by the four ethical principles of autonomy, beneficence, non-maleficence and justice.

Participants will be informed that their participation in this study is voluntary and that their confidentiality will be maintained throughout the study. Participants will also be assured that they are free to withdraw at any stage of the study and could opt-out of questions that they are not comfortable with. All identifying information will be removed. All electronic records will be accessed through a password-encrypted database that only the principal investigator has access to. No direct incentives will be issued to participants. Before initiating the self-administered questionnaires, informed consent forms will be signed by all study participants. A waiver of consent has been attained for the document review. Information sheets and consent forms will be translated into relevant local languages (isiXhosa, siSwati and isiZulu). They will also be assured that data collected will be used only for the purposes of the study.

Findings of the study will be disseminated widely to all stakeholders, including participants; and will be used to inform both provincial and national strategies to expand and sustain provision of high-quality cancer screening, diagnosis, treatment and palliative services, and promote community-based cancer care programmes. Results will be presented at annual partner meetings, national and international conferences. Results will also be published in open access peer-reviewed journals to facilitate broad access to findings.

**Author affiliations**
[1]Faculty of Health Sciences, Health Systems Enablement and Innovation, University of the Witwatersrand, Johannesburg, South Africa
[2]Economics, Nelson Mandela University, Gqeberha, South Africa
[3]Public Health, Walter Sisulu University, Mthatha, South Africa
[4]Oncology, Nelson Mandela Academic Hospital, Mthatha, South Africa
[5]Statistics, Walter Sisulu University, Mthatha, South Africa
[6]The George Institute for Global Health, University of New South Wales, Sydney, New South Wales, Australia
[7]School of Population Health, University of New South Wales, Sydney, New South Wales, Australia

**Acknowledgements** The authors would like to thank the support received from patient representatives and officials of the participating hospitals and provinces for their assistance.

**Contributors** WC conceived the research, sourced funding, engaged stakeholders, completed the first draft of the manuscript and jointly approved final draft. OM edited and commented on versions of the manuscript and incorporated and addressed feedback from the coauthors. SAM edited versions of the manuscript, provided methodological strategy, validated the quantitative survey tool and jointly approved final draft. ZJ is the content expert, edited and commented on versions of the manuscript. BS facilitated ethics and research access approvals, edited version of the manuscript. VE validated the quantitative survey tool and edited versions of the manuscript. DH, LG, NW, CZ, LB, SK, JN, SS, OG and AM edited the versions of the manuscript.

**Funding** This work was supported by the Bristol Myers Squibb Foundation grant number [1028].

**Competing interests** None declared.

**Patient and public involvement** Patients and/or the public were involved in the design, or conduct, or reporting or dissemination plans of this research. Refer to the Methods section for further details.

**Patient consent for publication** Not applicable.

**Provenance and peer review** Not commissioned; externally peer reviewed.

**ORCID iDs**
Onke R Mnyaka http://orcid.org/0000-0002-1834-6136
Buyiswa Swartbooi http://orcid.org/0000-0002-4536-1710
Sikhumbuzo A Mabunda http://orcid.org/0000-0001-9458-3742

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
