## [Reviewer comments · BMJ Open]

ARTICLE DETAILS

TITLE (PROVISIONAL)	A protocol of mixed-methods assessment of demographic, epidemiological, and clinical profile of decentralised cancer patients at Nelson Mandela Academic Hospital and Rob Ferreira Hospital, South Africa
AUTHORS	Chitha, Wezile; Jafta, Zukiswa; Mnyaka, Onke; Hongoro, Danleen; Godlimpi, Lizo; Swartbooi, Buyiswa; Williams, Natasha; Zungu, Christopher; Buthi, Lazola; Kuseni, Sibulelo; Nasila, John; Sibulawa, Siyabonga; Giwu, Olona; Mavimbela, Awam; Essel, Vivien; Mabunda, Sikhumbuzo

VERSION 1 – REVIEW

REVIEWER	Tod, Bianca Stellenbosch University, Dermatology
REVIEW RETURNED	16-Aug-2021

GENERAL COMMENTS	This is very necessary and important research. The researchers should be commended for undertaking this project. There are a few minor issues I would recommend attending to. It looks like a lot, but there are 2 studies here! Please bear in mind I am a clinician, not a public health specialist. GENERAL POINTS 1. You mention that Appendix A is validated- where/ by whom?2. It is not clear to me where you are in the decentralisation process? Chemo patients have been decentralised but this study provides baseline data in justifying decentralisation? One of your objectives is to report on decentralisation, which sounds like it is complete? It's not clear to me if decentralisation refers only to use of rural clinics, or use of NM and RF instead of EL and PTA? This issue should be clarified.3. Sub-study 1- data collection dates planned?4. Cancer patient registry and prospective follow up for sub-study 2- I think more information should be given. Does the registry include every cancer patient? Can patients opt out? How far back does it go? Do you have any idea how many patients will be included based on retrospective data? Has the registry been validated? How long will the prospective component last, if you are recruiting from 1 March 2019- 28 Feb 2022? How will their survival time be investigated?5. Will patients from sub-study 1 be linked to their data in sub-study 2?6. I suggest clarifying your triangulation of design concept (see 3, below).7. There are a few minor grammatical errors- the manuscript needs to be checked again. A few of the sentences are awkward
---

	and would benefit from revision. I find as an author these are easy to miss as you have read the manuscript so many times. SPECIFIC POINTS:  1. Abstract: line 38-42: sentence not clear. It reads as if the 424 patients are the only participants in the ambi-directional study. 2. Introduction: lines 72-78 & 82-93: too much information, I suggest significant editing. 3. Figure 1: This figure is not clear/ effective. I suggest leaving it out or reworking it. It may be a good opportunity to better explain your concept of triangulation of design.
--	---

REVIEWER	Weaver, Sallie J. National Cancer Institute
REVIEW RETURNED	08-Nov-2021

GENERAL COMMENTS	This paper describes the protocol for a descriptive study summarizing characteristics of a randomly sampled cohort of patients treated for cancer from 2019 through 2022 by two South African hospitals implementing a decentralized care delivery model for patients receiving chemotherapy. The protocol aims to describe demographic, epidemiological, and clinical characteristics using survey data and abstracted registry data. It is a reasonable fit for the BMJ Open protocol paper format, however several areas could be clarified in the study rationale and methods.  1. Premise: The goal to decentralize delivery of therapeutic treatment for cancer and enable high quality cancer care close to where patients live is laudable and important. However, the introduction provides limited insight regarding the role that care access issues (or degree to which)—broadly defined—play in the cancer incidence and death rates described at length in the background. It would be helpful for readers to know what work the participating hospitals/health systems did to understand access challenges, distances traveled, or the degree to which travel and other factors play roles in care delays or inability to receive guideline-concordant chemotherapy among populations in their catchment areas. I am guessing work of this type was done locally to inform the design of the decentralized care delivery model and it would be useful to summarize as part of the premise for this study. A few examples of travel time were described later in the description of the study setting—however, more systematic preliminary work was also likely done to provide an impetus for this study and could be described. 2. Premise and methods: The primary objective of this study was a bit muddy in my read. Could the authors clarify the relationship between this protocol/sub studies and the larger study that is mentioned? For example, the abstract described this protocol as “baseline” data, but it was not quite clear if the cohort enrolled in the survey and registry data abstraction have participated in the decentralized model, if they will participate at some future time, or if they are the baseline cohort who received current usual care. Clarification of these broader aims could help strengthen the contribution of this piece in my opinion. 3. Methods: Could the authors clarify how the decentralized care delivery model works? How it is staffed, etc.? It would be helpful to clarify if/how results from these analyses will be used to inform design and operations of the decentralized delivery model.
--

	4. Methods: The authors indicate the patient survey to be used in the study is “standardized and validated.” Please provide appropriate citations for survey development and testing. 5. Methods: Could the authors provide clarification about how survey and registry data will be matched, how disagreements between survey and registry responses will be adjudicated, and how targeted data elements will provide insight on issues that decentralized care models might address? (e.g., transportation, travel, and other broader access to care challenges, non-access related factors that influence receipt of timely, guideline concordant care) 6. Methods: Study limitations are mentioned briefly in a single bullet point strengths/limitations section, however, methods to address known limitations are not described in the methods section (e.g., how will data missingness be evaluated and addressed?). 7. The paper could benefit from additional discussion of the ambidirectional design depicted in Figure 1 would be useful to some readers who are not particularly familiar with such study designs. Alternatively, a figure note detailing the retrospective vs. prospective elements of design could be added to Figure 1.
--	--

VERSION 1 – AUTHOR RESPONSE

Reviewer: 1

Dr. Bianca Tod, Stellenbosch University

Comments to the Author:

This is very necessary and important research. The researchers should be commended for undertaking this project. There are a few minor issues I would recommend attending to. It looks like a lot, but there are 2 studies here! Please bear in mind I am a clinician, not a public health specialist.

GENERAL POINTS

1. You mention that Appendix A is validated- where/ by whom?

Author response: Thank you for the comment.

- The survey tool for patients has 38 questions developed through literature review and whose content validity was reviewed by three experts (two Public Health specialists and an Oncologist). The questionnaire asks about the patient demographic profile (questions 1-7) and Epidemiological and clinical profile of various cancers (questions 8-38).
- The survey tool was piloted in two hospitals, one in each province. This was done to test and ensure the effectiveness of the tool. There was never an intention to publish the results from the pilot study, rather to inform the design of the tool. Once the pilot study was done, all necessary adjustments were made to the data collection tool, thus ensuring that all questions will enhance the validity and reliability of the study findings.

2. It is not clear to me where you are in the decentralisation process? Chemo patients have been decentralised, but this study provides baseline data in justifying decentralisation? One of your objectives is to report on decentralisation, which sounds like it is complete? It's not clear to me if decentralisation refers only to use of rural clinics, or use of NM and RF instead of EL and PTA? This issue should be clarified.

Author response: Thank you for the comment.

- The current decentralised model of care is only limited to patients requiring chemotherapy due to

limited resources.

- The main research project/ study seeks to understand the existing cancer care delivery systems including looking at different decentralisation strategies and understanding system challenges/ difficulties experienced at different levels of the health system. Hopefully, this will help understand what is working and what does not work.
- The current study seeks to provide insight on the distribution and types of cancers in areas where there is currently an underestimation of the burden of disease and as a result incorrect understanding of the levels of risk within the local populations.
- As noted in the background of our manuscript, the two hospitals aim to establish centres of excellence in cancer care, a network of cancer care satellite sites at district hospital level and community-based cancer care services. The primary goal is to ensure that patients requiring cancer care are able to get their quality care closer to where they live, health professionals are able to screen and diagnose early, unnecessary delays to treatment are reduced and patients get quality palliative care closer to home. Secondary goals include reducing long distance travelled when seeking cancer care, reducing costs of seeking cancer care and reducing unnecessary and/or late referral of cancer patients.

3. Sub-study 1- data collection dates planned?

Author response: Thank you for the comment. Data collection is scheduled to commence during mid-January 2022 and at the end of September 2022.

4. Cancer patient registry and prospective follow up for sub-study 2- I think more information should be given.

- Does the registry include every cancer patient? Can patients opt out? How far back does it go?
- Do you have any idea how many patients will be included based on retrospective data? Has the registry been validated?
- How long will the prospective component last, if you are recruiting from 1 March 2019- 28 Feb 2022?
- How will their survival time be investigated?

Author response: Thank you for the comment

- All patients under the care of the unit at any stage between the 01st of March 2019 and the 28th of February 2022 will be included.
- Patients will be provided with study information sheet and sign informed consent form prior their participation in the study.

5. Will patients from sub-study 1 be linked to their data in sub-study 2?

Author response: Thank you for the comment. In sub-study 1 patients will be asked about their history of cancer in addition to their present cancer condition. The document review (sub-study 2) will capture both the retrospective and prospective data. The data collected from a document review (cancer registry) will be used to complement and/or verify the information that has been gathered through the survey questionnaire in sub-study 1.

6. I suggest clarifying your triangulation of design concept (see 3, below).

Author response: Thank you for the comment

7. There are a few minor grammatical errors- the manuscript needs to be checked again. A few of the sentences are awkward and would benefit from revision. I find as an author these are easy to miss as you have read the manuscript so many times.

Author response: Thank you for the comment, some sentences have been revised as advised.

SPECIFIC POINTS:

1. Abstract: line 38-42: sentence not clear. It reads as if the 424 patients are the only participants in

the ambi-directional study.

Author response: Thank you for the comment, the paragraphs have been revised.

2. Introduction: lines 72-78 & 82-93: too much information, I suggest significant editing.

Author response: Thank you for the comment, the paragraphs have been revised as advised.

3. Figure 1: This figure is not clear/ effective. I suggest leaving it out or reworking it. It may be a good opportunity to better explain your concept of triangulation of design.

Author response: Thank you for the comment.

Reviewer: 2

Dr. Sallie J. Weaver, National Cancer Institute

Comments to the Author:

This paper describes the protocol for a descriptive study summarizing characteristics of a randomly sampled cohort of patients treated for cancer from 2019 through 2022 by two South African hospitals implementing a decentralized care delivery model for patients receiving chemotherapy. The protocol aims to describe demographic, epidemiological, and clinical characteristics using survey data and abstracted registry data. It is a reasonable fit for the BMJ Open protocol paper format, however several areas could be clarified in the study rationale and methods.

1. Premise: The goal to decentralize delivery of therapeutic treatment for cancer and enable high quality cancer care close to where patients live is laudable and important. However, the introduction provides limited insight regarding the role that care access issues (or degree to which)—broadly defined—play in the cancer incidence and death rates described at length in the background. It would be helpful for readers to know what work the participating hospitals/health systems did to understand access challenges, distances travelled, or the degree to which travel and other factors play roles in care delays or inability to receive guideline-concordant chemotherapy among populations in their catchment areas. I am guessing work of this type was done locally to inform the design of the decentralized care delivery model and it would be useful to summarize as part of the premise for this study. A few examples of travel time were described later in the description of the study setting—however, more systematic preliminary work was also likely done to provide an impetus for this study and could be described.

Author response: Thank you for the comment

2. Premise and methods: The primary objective of this study was a bit muddy in my read. Could the authors clarify the relationship between this protocol/sub studies and the larger study that is mentioned?

For example, the abstract described this protocol as “baseline” data, but it was not quite clear if the cohort enrolled in the survey and registry data abstraction have participated in the decentralized model, if they will participate at some future time, or if they are the baseline cohort who received current usual care. Clarification of these broader aims could help strengthen the contribution of this piece in my opinion.

Author response: Thank you for the comment.

- The period (1 March 2019- 28 Feb 2022) under investigation is in alignment with the implementation decentralisation of cancer care services in the two hospitals. As indicated in the background of our manuscript, the Nelson Mandela Academic and Rob Ferreira hospitals embarked on a decentralised model of cancer care delivery from 01 March 2019.

- The main research project/ study seeks to understand the existing cancer care delivery systems including looking at different decentralisation strategies and understanding system challenges/ difficulties experienced at different levels of the health system. Hopefully, this will help understand what is working and what does not work.

- The current study seeks to provide insight on the distribution and types of cancers in areas where

there is currently an underestimation of the burden of disease and as a result incorrect understanding of the levels of risk within the local populations.

- Information will be extracted from the patient registry to respond to the questions on the extraction tool. All patients under the care of the unit at any stage between the 01st of March 2019 and the 28th of February 2022 will be included.

3. Methods: Could the authors clarify how the decentralized care delivery model works? How it is staffed, etc.? It would be helpful to clarify if/how results from these analyses will be used to inform design and operations of the decentralized delivery model.

Author response: Thank you for the comment

- The current decentralised model of care is only limited to patients requiring chemotherapy due to limited resources including healthcare staff.

- It is envisaged that this study will provide insight on the distribution and types of cancers in areas where there is currently an underestimation of the burden of disease and as a result incorrect understanding of the levels of risk within the local populations. Moreover, establish the extent of the problem in both health facilities and possibly justify the need for decentralisation of cancer care services and help inform cancer preventive strategies in South Africa and other similar settings.

4. Methods: The authors indicate the patient survey to be used in the study is “standardized and validated.” Please provide appropriate citations for survey development and testing.

Author response: Thank you for the comment.

- The survey tool for patients has 38 questions developed through literature review and whose content validity was reviewed by three experts (two Public Health specialists and an Oncologist).

The questionnaire asks about the patient demographic profile (questions 1-7) and Epidemiological and clinical profile of various cancers (questions 8-38).

- The survey tool was piloted in two hospitals, that is, one hospital from each of the selected study regions. This was done to test and ensure the effectiveness of the data collection tool. There was never an intention to publish the results from the pilot results as they were only used to inform the design data collection tool. Once the pilot study was done, all necessary adjustments to the data collection tool were made thus ensuring that all questions will enhance the validity and reliability of the study findings.

5. Methods: Could the authors provide clarification about how survey and registry data will be matched, how disagreements between survey and registry responses will be adjudicated, and how targeted data elements will provide insight on issues that decentralized care models might address? (e.g., transportation, travel, and other broader access to care challenges, non-access related factors that influence receipt of timely, guideline concordant care).

Author response: Thank you for the comment.

- This approach will also compensate for the potential limitations of a single data collection method and to triangulate the data as a means of checking the consistency of the study findings.

- It is believed that information on socio-demographic characteristics of patients, geographic distribution of cancers, comorbid conditions of cancer patients, etc. is pivotal in understanding challenges/ difficulties experienced by patients at different levels within the health system. For example, patients who are geographically located in deep rural areas where there is generally poor infrastructure development (e.g., roads, health facilities, etc.) are more likely to be confronted with difficulties to have timeous access cancer care services and thus present late for diagnosis, etc. Therefore, understanding their conditions might provide useful information that can be used to advocate for a variety of decentralisation strategies.

6. Methods: Study limitations are mentioned briefly in a single bullet point strengths/limitations section, however, methods to address known limitations are not described in the methods section (e.g., how will data missingness be evaluated and addressed?).

Author response: Thank you for the comment.

7. The paper could benefit from additional discussion of the ambi-directional design depicted in Figure 1 would be useful to some readers who are not particularly familiar with such study designs. Alternatively, a figure note detailing the retrospective vs. prospective elements of design could be added to Figure 1.

Author response: Thank you for the comment.

VERSION 2 – REVIEW

REVIEWER	Tod, Bianca Stellenbosch University, Dermatology
REVIEW RETURNED	12-Jan-2022

GENERAL COMMENTS	Thank you for your edited manuscript. In my opinion, it has improved significantly. There are just 3 points that are still not clear to me (and should be included in the manuscript please).  1. (Point 4 in my previous review) Please include more details regarding the registry used for sub-study 2: Is this a formal, validated cancer registry, or simply a clinic log book? 2. (Point 4 in my previous review) Please expand on how you will determine survival times- you mention reviewing death certificates in the limitations section, will you check all the patients' ID numbers, for example? Will you phone patients or their families? 3. (Point 5 in my previous review) I am still not clear on the linkage of individual patients in sub-study 1 and sub-study 2, if the data is de-identified.
---

REVIEWER	Weaver, Sallie J. National Cancer Institute
REVIEW RETURNED	12-Jan-2022

GENERAL COMMENTS	Thank you for the opportunity to review this protocol manuscript. I appreciate the responsive revisions and replies to prior reviewer feedback. This revision is mostly responsive to key feedback to clarify the definition and details of “decentralization” in this study and to address limitations. I do offer additional comments below.  1. Thank you for clarifying that this protocol is part of a larger body of work in your reply letter. Please include a sentence or two to the manuscript indicating that this protocol is a piece of a larger body of work as well. It was not clear to me where this was described in the revised manuscript. 2. Thank you for describing the decentralization process and details on pg. 6 and 7 of the revised manuscript. It might be useful to also include a figure or timeline that displays the decentralization process/timeline, as well as key data collection points for this study protocol. Perhaps an easy option would be to add the decentralization information to existing Figure 1. Please also clarify if comparisons between participants visiting decentralized sites vs. tertiary sites will be considered. 3. I appreciate the additional information about the development of the survey to be used in sub-study 1. However, I would be
---

	cautious about using language overstating that it is a “validated” or “valid and reliable” survey when only some aspects of content validity/face validity have been pilot tested. Additionally, the description of expert ratings of survey items on pg. 12 indicates their ratings appear in Appendix B, however, this is not the case. The information in the document titled Appendix B attached to the revision included the document review template.
--	---

VERSION 2 – AUTHOR RESPONSE

Reviewer 1:

Dr. Bianca Tod, Stellenbosch University

Comments to the Author:

Thank you for your edited manuscript. In my opinion, it has improved significantly. There are just 3 points that are still not clear to me (and should be included in the manuscript please).

1. (Point 4 in my previous review) Please include more details regarding the registry used for sub-study 2: Is this a formal, validated cancer registry, or simply a clinic logbook?

Author response: Thank you for the comment. The registry is based on information we need for our study objectives, (demographic characteristics, prevalent cancers and their epidemiology, duration of diagnosis/treatment and outcome (death). This registry was piloted and found to be appropriate for our study. The registry has not been validated. It is similar to a clinic logbook, but the registry is in an electronic form. This has been added in the main document, line 185-186 and 234-235.

2. (Point 4 in my previous review) Please expand on how you will determine survival times- you mention reviewing death certificates in the limitations section, will you check all the patients' ID numbers, for example? Will you phone patients or their families?

Author response: The researchers will only have access to the units that are depicted in Table 1. The hospital staff on the other hand will have access to the units that are shown in Table 2. When the researchers need clarity on the information which is captured on Table 1, they will communicate with the relevant hospital staff member who has access to the full clinical record of the specific patient including if he/she died in the hospital as they will have access to the BI1663. Additionally, the hospital staff are also able to communicate with families of the patients since they are dealing with a small population of patients. See Table 1 and Table 2 below.

Table 1: what is seen by the research team.

Subject number Date of birth Sex Cancer type ICD10 code Date of cancer diagnosis Date of death ...

Table 2:

Subject number Name Surname Patient ID number Folder number Sex Cancer type ICD10 code Date of cancer diagnosis Date of death ...

3. (Point 5 in my previous review) I am still not clear on the linkage of individual patients in sub-study 1 and sub-study 2, if the data is de-identified.

Author response: Sub-study 1 will be done independently of sub-study 2; as such, individual patients in these sub-studies will not be linked. It is possible that some of the same patients will be recruited for both sub-studies; however, the goal is not to link them in our study.

Reviewer: 2

Dr. Sallie J. Weaver, National Cancer Institute

Comments to the Author:

Thank you for the opportunity to review this protocol manuscript. I appreciate the responsive revisions

and replies to prior reviewer feedback. This revision is mostly responsive to key feedback to clarify the definition and details of “decentralization” in this study and to address limitations. I do offer additional comments below.

1. Thank you for clarifying that this protocol is part of a larger body of work in your reply letter. Please include a sentence or two to the manuscript indicating that this protocol is a piece of a larger body of work as well. It was not clear to me where this was described in the revised manuscript.

Author response: Thank you for the comment. This is mentioned in the main document under the methods section, line number 173-176.

2. Thank you for describing the decentralization process and details on pg. 6 and 7 of the revised manuscript. It might be useful to also include a figure or timeline that displays the decentralization process/timeline, as well as key data collection points for this study protocol. Perhaps an easy option would be to add the decentralization information to existing Figure 1. Please also clarify if comparisons between participants visiting decentralized sites vs. tertiary sites will be considered.

Author response: Thank you for the comment.

a) Below is the timeline and a brief description of the current decentralisation process. The three hospitals have implemented the first phase (decentralisation of chemotherapy services) of the decentralisation process at different times. Figure 1 summarises the timeline of the decentralisation process. Figure 1 has been added and cited in the main document, line 127-128 and 391-394. Figure 1 which was titled “Summary of ambi-directional cohort sub-study” has now been revised to be figure 2 and this information has also been revised in the main document, lines 188 and 396.

b) On the current study we will not be doing any comparisons. However, there is a possibility of conducting an independent study that will focus on comparisons between participants visiting decentralized sites vs. tertiary sites at a later stage.

3. I appreciate the additional information about the development of the survey to be used in sub-study 1.

a. However, I would be cautious about using language overstating that it is a “validated” or “valid and reliable” survey when only some aspects of content validity/face validity have been pilot tested.

Author response: Thank you for the comment. The tool used for sub-study 1 was validated in full and we only limited the use of those words to sub-study 1.

b. Additionally, the description of expert ratings of survey items on pg. 12 indicates their ratings appear in Appendix B, however, this is not the case. The information in the document titled Appendix B attached to the revision included the document review template.

Author response: Thank you for the comment. This has been rectified. A separate appendix (appendix C) has been included as part of the supplementary documents.

VERSION 3 – REVIEW

REVIEWER	Tod, Bianca Stellenbosch University, Dermatology
REVIEW RETURNED	09-Mar-2022
GENERAL COMMENTS	Thanks for your clarification. I am satisfied with your changes and comments.